# Mapping organic carbon vulnerable to mobile bottom fishing in currently unfished areas of the Norwegian continental margin

Markus Diesing,[1] Marija Sciberras,[2] Terje Thorsnes,[1] Lilja Rún Bjarnadóttir,[1] and Øyvind Grøner Moe[3]

[1]Geological Survey of Norway, Trondheim, Norway
[2] Lyell Centre, Heriot-Watt University, Edinburgh, United Kingdom
[3] Directorate of Fisheries, Bergen, Norway

*Correspondence to*: Markus Diesing (markus.diesing@ngu.no)

**Abstract.** Organic carbon stored in continental margin sediments might be at risk by widespread mobile bottom fishing, potentially leading to reductions of organic carbon stocks, increased ocean acidification, additional atmospheric carbon dioxide emissions and a reduction of the buffering capacity of the ocean. Spatially explicit studies that have been conducted to inform marine management have so far looked at organic carbon stocks that have already been affected by mobile bottom fishing. Here, we focus instead on areas on the Norwegian continental margin that are currently not fished, based on fishing data covering the years 2009 – 2020. Using these data and spatial prediction methods, we estimate that the surface sediment layer (0 – 2 cm) in unfished areas covering 765,600 km² contains 139.2 Tg of organic carbon. Based on data from a meta-analysis of demersal fishing impacts on organic carbon density and estimated reductions in sediment thickness due to fishing-induced erosion, we estimate that 18.7 Tg (1.9 – 33.5 Tg) of organic carbon might be lost due to mobile bottom fishing in a scenario where each grid cell is fished evenly over the entire area and down to the full depth of the surface layer. Approximately one third of this vulnerable organic carbon is currently located in existing area-based protection measures. Additional protection could be guided by hotspots of vulnerable organic carbon, which are mainly found in the Barents Sea. We argue that the protection of vulnerable organic carbon that is at high risk of being lost e.g. in areas becoming accessible to fishing due to sea ice retreat such as in the northern Barents Sea should be given a high priority.

## 1 Introduction

Continental margin sediments are a major hotspot for organic carbon burial (Burdige, 2007) and thus play a vital role in the carbon cycle. Disturbance of sedimentary organic carbon stored in margin sediments by mobile bottom fishing might lead to reductions of the organic carbon stocks (Zhang et al., 2024), increased ocean acidification and atmospheric $CO_2$ emissions (Atwood et al., 2024), and a reduced buffering capacity of the ocean (Sala et al., 2021). Although the potential impacts of mobile bottom fishing on seabed biogeochemistry and the marine carbon cycle have been a matter of research since at least the beginning of the 1990s (Mayer et al., 1991), it is only recently that the interest in this topic has markedly increased. This can mainly be attributed to a global study that estimated aqueous emissions of 0.58 – 1.47 Pg $CO_2$ yr$^{-1}$ due to mobile bottom

fishing (Sala et al., 2021). A subsequent study estimated that between 1996 and 2020, mobile bottom fishing might have emitted $0.34 - 0.37$ Pg $CO_2$ yr$^{-1}$ to the atmosphere globally (Atwood et al., 2024). Such high estimates have been called into question (Hiddink et al., 2023; Smeaton and Austin, 2022; Zhang et al., 2024), but certainly also garnered interest of the media, stakeholders, and environmental non-government organisations in the topic.

Field studies concerning the impact of mobile bottom fishing on organic carbon in seabed sediments have been conducted with different study designs (experimental, comparative control-impact and comparative gradient studies), fishing gear types and under differing environmental conditions (Tiano et al., 2024). A review of 49 published studies which directly investigated the impact of mobile bottom fishing on organic carbon had mixed results: No significant effect was reported by 61% of the studies, 29% reported lower organic carbon contents and 10% reported higher organic carbon contents attributed to fishing (Epstein et al., 2022). This can be attributed to the many mechanisms that impact organic carbon stocks in positive and negative ways (Porz et al., 2024): Sediment resuspension will likely lead to a reduction of organic carbon locally (De Borger et al., 2021; Bradshaw et al., 2021; Morys et al., 2021; Paradis et al., 2021; van de Velde et al., 2018), while the transport and deposition in deeper areas could increase stocks farther away (Paradis et al., 2019, 2022; Porz et al., 2024). Churning of the sediment by penetrating gear components likely increases sediment mixing and porewater fluxes, with opposite impacts on organic carbon stocks (De Borger et al., 2021; Bunke et al., 2019; Duplisea et al., 2001; Oberle et al., 2016a; Paradis et al., 2021; van de Velde et al., 2018). Increased mortality of animals living in and on the sediment will likely lead to reduced remineralisation of organic carbon and bioturbation of the sediment, again with opposite effects on stocks (Tiano et al., 2019). An increased resuspension of nutrients into the water column might increase primary production in surface waters (Dounas et al., 2007), while increased turbidity would have the opposite effect (Porz et al., 2024). Accounting for all these and more effects in experimental or modelling studies remains a challenge. A clearer pattern of the overall effects has, however, emerged since the publication of a global meta-analysis of experimental and comparative studies (Tiano et al., 2024): The analysis of experimental and comparative control-impact studies showed that fishing disturbance significantly reduced the content of chlorophyll-a, phaeopigments and proteins, which are indicators of labile organic carbon. Effects on these compounds were more pronounced in surface sediments ($0 - 2$ cm), where the impact on total organic carbon content also became significant (Tiano et al., 2024). Several studies have attempted to translate the current understanding of the impacts of mobile bottom fishing and seabed organic carbon stocks into spatial analyses that can guide marine spatial management. Black et al. (2022) assessed the potential vulnerability of sedimentary organic carbon stocks to bottom trawling and dredging related disturbances within the exclusive economic zone of the United Kingdom. They based their approach largely on the study by Sala et al. (2021) but instead of estimating aqueous $CO_2$ emissions they developed a carbon vulnerability ranking. Another study from the United Kingdom identified priority areas to manage mobile bottom fishing and estimated a cumulative disturbance of organic carbon by fishing of 109 Tg yr$^{-1}$ (Epstein and Roberts, 2022). Areas with both high organic carbon stocks and disturbance were found to be geographically restricted thus enabling the identification of potential priority areas for precautionary carbon management or future research. More recently, Porz et al. (2024) quantified mobile bottom fishing impacts on sedimentary organic carbon stocks in the North Sea with a 3D-coupled numerical model. They simulated six years with different spatial distribution of

fishing impact, including a baseline scenario, a scenario without any bottom fishing and four management scenarios. The authors found that North Sea sediments contained 0.55 Tg less organic carbon by the end of each year when comparing fished simulations with the unfished simulation, equalling aqueous emissions of 2.0 Tg $CO_2$ each year. Results showed high spatial variability of the impact, with a high loss of organic carbon in some areas, while organic carbon stocks increased in nearby unfished areas due to transport and redeposition. The largest positive management effects arose from fishing closures in areas where organic carbon is both plentiful and labile. Another modelling study focussing on the North Sea found that gain and loss of sedimentary organic carbon may occur in weakly fished areas, whereas long-term carbon storage is reduced in intensively fished areas (Zhang et al., 2024). By upscaling to the global ocean, the authors found that their estimates were less than 10% of the previous estimate made by Sala et al. (2021). Zhang et al. (2024) attributed this large discrepancy to the redeposition of resuspended organic carbon and the impact of bottom fishing on benthic fauna, which partly offset losses of organic carbon and were not considered by Sala et al. (2021).

The above-described studies (Black et al., 2022; Epstein and Roberts, 2022; Porz et al., 2024; Zhang et al., 2024) and those of Sala et al. (2021) and Atwood et al. (2024) all have focussed on seabed areas impacted by mobile bottom fishing for obvious reasons. The footprint of fishing is, however, not stable over time (Watson and Tidd, 2018) and might change in the future. Mobile bottom fishing is operating in increasingly deeper water (Watson and Morato, 2013). Trawling penetrates rapidly into Arctic shelf areas previously protected by extensive ice-cover as a response to interannual sea ice loss (Fauchald et al., 2021). Without management, global fishing fleets are expected to shift poleward by the end of the century, driven by polar fishing gears moving to higher Arctic areas and tropical fishing gears expanding both within the tropics and poleward (Cruz et al., 2024). Mobile bottom fishing could hence move into currently unimpacted or even pristine areas (e.g., where sea-ice is retreating), but the potential impacts on organic carbon stocks in those areas have so far not been investigated.

The objective of this study is therefore to identify and quantify organic carbon stocks in currently unfished areas that would become vulnerable to loss by mobile bottom fishing on the Norwegian continental margin should the fishing distribution change. In the context of this study, loss means the reduction of local organic carbon stocks. We carry out a meta-analysis of experimental control-impact studies that record changes in both organic carbon content and dry bulk density following a fishing event (referred thereafter to as acute impact studies). Additionally, we estimate reductions in sediment thickness due to fishing-induced erosion. We subsequently apply those results to spatially predicted organic carbon stocks in currently unfished areas. We focus on the surface layer (0 – 2 cm) because the largest and most significant effects from mobile bottom fishing were observed in that layer (Tiano et al., 2024) and fishing gear used in Norway on average does not penetrate much deeper than the surface layer (Hiddink et al., 2017). We also compare our results with existing area-based protection measures to estimate the level of protection afforded to vulnerable organic carbon.

## 2 Methods

### 2.1 Study site

Our study site (Figure 1a) comprises the Norwegian continental shelf, slope and the shallowest part of the abyss. The landward limit is the baseline relative to which maritime zones such as the territorial sea and the exclusive economic zone are defined. The seaward limit was placed at 50 km distance from the seaward boundary of the continental slope to make best use of the available data. The extent of the continental shelf and slope were based on Harris et al. (2014).

The continental shelf west off Svalbard has been closed to mobile bottom fishing since the early 1980s under the harvest regulations (https://lovdata.no/dokument/SF/forskrift/2021-12-23-3910) (Figure 1b). Seabed areas below 1,000 m water depth have been protected from the impacts of bottom fishing in Norway since 2011 according to the Regulation on vulnerable marine ecosystems (https://lovdata.no/dokument/SF/forskrift/2011-07-01-755). This regulation was amended in 2019 to respond to rapid global warming in the northern Barents Sea, leading to the protection of 442,022 $km^2$ of seabed in previously sea ice-covered areas and ten smaller protection zones in historically fished seabed areas (Jørgensen et al., 2020). These protection measures specifically aim at the protection of vulnerable marine ecosystems. Additionally, several smaller protection zones have been established with the aim to protect representative cold-water coral reefs against the impacts of bottom-contacting fishing gear (https://lovdata.no/dokument/SF/forskrift/2016-01-08-8).

### 2.2 Definitions

Given inconsistencies in the literature we first provide some definitions of terms we use in this study to provide more clarity: Dry bulk density refers to the ratio between the mass of dry solids (sediment grains) and the volume of the water-saturated sediment sample measured in $kg\ m^{-3}$ or similar (Flemming and Delafontaine, 2016). Organic carbon content is the mass of organic carbon per unit mass of dry sediment commonly measured as $g\ kg^{-1}$ or weight-%. Note that the latter is strictly speaking not correct, as the content is relating to a mass not weight. Due to the widespread use of weight-% as a unit for organic carbon content, we also make use of this term. The product of organic carbon content and dry bulk density is the organic carbon density, measured in $kg\ m^{-3}$. Alternatively, this could also be seen as a concentration (Flemming and Delafontaine, 2000, 2016). However, since the term organic carbon concentration is frequently used interchangeably with organic carbon content, we refrain from using it in this study. Organic carbon stock is the mass of organic carbon per area in a defined depth interval within the sediment. It can be derived by multiplying organic carbon density with the thickness of the sediment under consideration. Organic carbon stocks are typically reported as $kg\ C\ m^{-2}$ but note that it is important to refer to the sediment depth interval for which the stock has been derived. Organic carbon reservoir is the mass of organic carbon in an area and a defined depth interval. Typically, reservoirs are calculated for larger areas such as a sea basin and expressed as Tg C or similar.

## 2.3 Method overview

The workflow of this study is summarised in Figure 2. In the first instance, we model and predict the spatial distribution of organic carbon content and dry bulk density in surface sediments and use these predictions to calculate the surface organic carbon stocks in the surface layer (0 – 2 cm). We then calculate surface swept volume ratios (SVR) from publicly available fishing intensity data (swept area ratio) and average gear penetration depths. The mass of disturbed organic carbon is calculated for areas where fishing is present (i.e., SVR > 0 yr$^{-1}$) and the annual cumulative disturbance of organic carbon stocks (Epstein and Roberts, 2022) is estimated. The mass of undisturbed organic carbon is estimated for areas without fishing activity (i.e., SVR = 0 yr$^{-1}$). Based on a meta-analysis of acute fishing impact studies and changes in sediment thickness due to resuspension of sediment, we estimate the mass of vulnerable organic carbon.

## 2.4 Spatial prediction of organic carbon content and dry bulk density

The methodology used to model and spatially predict organic carbon content and dry bulk density is an updated version of the one described in Diesing et al. (2024). We used the same response data and predictor variables (Diesing, 2024a, b) as in Diesing et al. (2024). The predictor variables were obtained as a raster stack and cropped to the study site. Seabed areas dominated by rock and boulders were excluded from further analysis. The datasets of the two response variables organic carbon content and dry bulk density were filtered to only include records between 0 cm and 2 cm depth below the sediment-water interface. The response data were averaged in cases where more than one value was falling into a grid cell of the predictor stack.

Modelling and spatial prediction was carried out with the quantile regression forest (QRF) algorithm (Meinshausen, 2006), which can be seen as an extension of the random forest (RF) algorithm (Breiman, 2001). Both QRF and RF are ensemble techniques that grow many regression trees and aggregate the predictions. While RF only outputs the conditional mean, QRF also returns the whole conditional distribution, based on which other measures of central tendency (e.g., median) and of prediction uncertainty can be obtained or calculated. Here we use the median, as the conditional distributions are most likely non-normal, and a median is not affected by extreme outliers.

We limited the predictor variables to a meaningful subset that was used for modelling with the forward feature (variable) selection algorithm implemented in the R package CAST (Meyer et al., 2024). Modelling was conducted with the k-fold nearest neighbour distance matching algorithm (Linnenbrink et al., 2024) that allowed us to estimate model performance with spatially separated folds. In this way, inflated performance estimates due to spatial autocorrelation in the response data were avoided. The performance of the final models was assessed based on the mean error (which measures bias), r-squared (which measures the explained variance) and the root mean squared error (which measures accuracy). We estimated the area of applicability (AOA) of both models using the aoa function (Meyer and Pebesma, 2021) of the R package CAST (Meyer et al., 2024). The AOA is a visual representation of the area where the combination of predictor variables is similar to what the model has been trained with.

## 2.5 Calculation of organic carbon stocks and reservoir sizes

Organic carbon stocks (OCS) are calculated by multiplying the predicted organic carbon contents (G) with the predicted dry bulk densities ($\rho_d$) and sediment thickness (d = 0.02 m):

$$OCS \ (kg \ m^{-2}) = \ G \ (-) \cdot \rho_d \ (kg \ m^{-3}) \cdot d(m) \tag{1}$$

The total reservoir size $m_{oc}$ was calculated by summing OCS of all pixels and multiplying with the area of one pixel (A = 16,000,000 m$^2$):

$$m_{OC} \ (Tg) = (A \ (m^2) \cdot \sum OCS \ (kg \ m^{-2}))/1,000,000,000 \tag{2}$$

## 2.6 Processing of fishing intensity data

Spatial data on fishing intensity were obtained from ICES (2021). The metric of interest in this context is the swept area ratio
(SAR), which is the cumulative area contacted by a fishing gear within a grid cell over one year divided by the surface area of the grid cell. Grid cells have the size of 0.05° by 0.05°. SAR was provided for the surface (0 – 2 cm) and subsurface (> 2 cm) layer, but only the former was used here as we are focussing on impacts in the surface layer. SAR estimates were provided for four higher-level metier groupings (beam trawl, dredge, demersal seine and otter trawl). The mean annual surface SAR was calculated separately for beam trawls, demersal seines and otter trawls using fishing intensity data for the period 2009 to 2020.
Dredges were not present in the study area, and therefore, were excluded from further analysis.

Based on the SAR estimates, surface swept volume ratios (SVR) were calculated by multiplying SAR with mean gear penetration depths (averaged across gear width). Beam and otter trawls have estimated mean gear penetration depths of 2.72 cm and 2.44 cm, respectively (Hiddink et al., 2017). This means that such gears penetrate the whole surface layer (0 – 2 cm). A value of 0.5 cm was used for seines, which is a conservative estimate provided by Epstein and Roberts (2022) based on Eigaard
et al. (2016). We consider all grid cells with a surface SVR > 0 yr$^{-1}$ as fished. Similarly, grid cells with a surface SVR = 0 yr$^{-1}$ are considered unfished.

## 2.7 Annual cumulative disturbance of organic carbon stocks in fished areas

We follow the methodology outlined in Epstein and Roberts (2022). The annual cumulative disturbance of seabed organic carbon stocks was calculated by multiplying the organic carbon stock by the SVR, which creates a quantitative value (kg C m$^{-2}$ yr$^{-1}$)
that indicates the annual level of disturbance on organic carbon stocks.

## 2.8 Meta-analysis

We carried out a meta-analysis based on data from the Demersal fishery Impacts on Sedimentary Organic Matter (DISOM) database (Paradis et al., 2024), supplemented with data from a systematic literature search described in Felgate et al. (2024). Given the available data, we restrict our study to fishing impacts on bulk organic carbon and make no attempt to account for different reactivity fractions. The analysis was therefore restricted to studies measuring changes in organic carbon content and dry bulk density in the surface layer (0 – 2 cm) within the first week of the fishing event taking place (acute effects). Organic carbon content (mass organic carbon per mass dry sediment measured in weight-% or similar) and dry bulk density (mass dry sediment per volume wet sediment measured in g cm$^{-3}$ or equivalent) are related to each other (Leipe et al., 2011), although the exact form of the relationship is site dependent. Therefore, only studies that reported organic carbon content and either dry bulk density, porosity or absolute water content were considered. The latter two were included as dry bulk density can be calculated from porosity or water content (Flemming and Delafontaine, 2000) with a high degree of accuracy. Organic carbon densities were calculated from organic carbon content and dry bulk density (Eq. 3) and subsequently analysed. Mean, standard deviation and sample size values for organic carbon density in fished and control (unfished) areas were used to estimate the log response ratio (lnRR), which quantifies the proportional change in organic carbon density due to demersal fishing (Hedges et al., 1999). For studies which did not provide standard deviation values, these were imputed as described in Tiano et al. (2024), and for studies with sample size of 1, the highest reported standard deviation for that response variable was assigned. A random effects model with restricted maximum likelihood estimator was used to calculate the mean response and the upper and lower 95% confidence intervals that describe the overall effect of fishing disturbance on organic carbon densities across studies (Viechtbauer, 2010). For ease of interpretation, we quote lnRR values in the results section as percent values, calculated as $(e^{lnRR} - 1) \cdot 100$.

We compared environmental conditions of the included studies of the meta-analysis with those of the Norwegian margin. To that end, we downloaded global data on water depth, mean sea surface primary productivity, and mean bottom current velocity from Bio-ORACLE v.2.2 (Assis et al., 2018). These environmental variables were found to significantly influence the effect of fishing disturbance on sedimentological and biogeochemical parameters (Tiano et al., 2024). We extracted values of those layers for the reported locations of the fishing impact studies and for 1,000 random locations within unfished areas on the Norwegian continental margin. We then plotted density curves for every environmental variable to visually check to what degree the curves of the fishing impact studies and the Norwegian margin overlapped (Figure S1).

## 2.9 Vulnerable organic carbon in unfished areas

Here we define vulnerable organic carbon as the carbon that might be lost due to a fishing event in a currently unfished location. More specifically, we consider vulnerable organic carbon as the organic carbon that would be lost due to fishing the entire area of a grid cell evenly over the full depth of the surface layer (i.e., SVR = 1 yr$^{-1}$). Assuming an even distribution of SVR allows us to identify spatial patterns and hotspots of vulnerable organic carbon in unfished areas. We acknowledge that areas

that are currently fished might also contain organic carbon vulnerable to fishing disturbance. However, this is not the focus of our research.

The product of organic carbon content and dry bulk density is also known as organic carbon density:

$$\rho_{OC} = G \cdot \rho_d \tag{3}$$

We can thus rearrange Eq. 1 as follows:


$$OCS = \rho_{OC} \cdot d \tag{4}$$

The organic carbon stock post disturbance $OCS_{pd}$ equates to:

$$OCS_{pd} = (\rho_{OC} + \delta\rho_{OC}) \cdot (d + \delta d) \tag{5}$$

The Greek letter δ symbolises changes to the organic carbon densities and sediment thickness, respectively. We can substitute:

$$\delta\rho_{OC} = \rho_{OC} \cdot RR_{\rho_{OC}} \tag{6}$$


with $RR_{\rho_{OC}}$ being the proportional change in organic carbon density as a result of fishing calculated from the meta-analysis. It follows:

$$OCS_{pd} = \left(\rho_{OC} + \rho_{OC} \cdot RR_{\rho_{OC}}\right) \cdot (d + \delta d) \tag{7}$$

$$OCS_{pd} = \rho_{OC}\left(1 + RR_{\rho_{OC}}\right) \cdot (d + \delta d) \tag{8}$$

Besides changes in organic carbon density, mobile bottom fishing also leads to erosion of sediment and hence a reduction of the sediment volume under consideration. Changes in sediment thickness act as a scaling factor that accounts for this erosion of the seabed. They can be estimated from the resuspended sediment mass per unit fished area $m$ (g m$^{-2}$), the proportion of the

sediment that resettles after resuspension $P_{crd}$ (dimensionless fraction) and the dry bulk density $\rho_d$ of the sediment:

$$\delta d = -m \cdot (1 - P_{crd})/\rho_d \tag{9}$$

The mass of the resuspended sediment might be estimated based on empirical relationships with the silt-clay content (weight-

%) as provided by Oberle et al. (2016b). These authors fitted linear functions to include 95% of all available data points thereby

creating an upper and lower linear limit as well as their mean. The proportion of the sediment that resettles after resuspension $P_{crd}$ was taken from Sala et al. (2021). Their value of $P_{crd} = 0.87$ was estimated using the average from a limited number of studies quantifying the amount of sediment lost following mobile bottom fishing or mining. While this is a current best estimate, it must be seen as associated with substantial uncertainty. We therefore conducted a sensitivity analysis to

estimate the impact of this parameter on vulnerable organic carbon (Table S1). By setting $P_{crd} = 1$ (all resuspended sediment resettles), $\delta d$ becomes 0 (eq. 9) and the mass of vulnerable organic carbon due to changes in organic carbon density ($\delta \rho_{OC}$) can be estimated separately.

The vulnerable organic carbon stock can finally be derived by subtracting the organic carbon stock post disturbance from the organic carbon stock prior to disturbance. To account for uncertainty in the analysis, we provide estimates for three scenarios:

The low impact scenario is based on the organic carbon stock calculated from the upper bound of the 95% confidence interval of $RR\rho_{oc}$ calculated from the meta-analysis and the lower linear limit of m calculated from Oberle et al. (2016b). Likewise, the high impact scenario uses the lower bound of $RR\rho_{oc}$ and the upper linear limit of m. Finally, the mean impact scenario is based on the mean response of the meta-analysis and the mean linear function from Oberle et al. (2016b).

Hotspots and coldspots of vulnerable organic carbon were identified with the Hot Spot Analysis (Getis-Ord Gi$^*$) tool in ArcGIS

10.8.2, which identifies statistically significant spatial clusters of high values (hotspot) and low values (coldspot). The vulnerable organic carbon map was converted from raster to point feature format prior to the analysis. Inverse distance was used for the conceptualisation of the spatial relationships and distances were measured as Euclidean distance. The resulting point feature class was converted to a raster file.

## 3 Results

### 3.1 Organic carbon stocks and reservoir size in the surface layer

The organic carbon content model was based on 685 observations. It had a mean error of 0.012 weight-%, a root mean squared error of 0.309 weight-%, an explained variance of 81.4% and an area of applicability equal to 95.7% of the total area. The dry bulk density model was based on 376 observations. It had a mean error of 0.100 g cm$^{-3}$, a root mean squared error of 0.166 g cm$^{-3}$, an explained variance of 75.1% and an area of applicability equal to 93.4% of the total area. Both models had low mean

errors close to zero, indicating that they were nearly unbiased. Explained variances and root mean squared errors were comparable to previous work (Diesing et al., 2024). The models were applicable in approximately 95% of the total area. All further analyses were therefore carried out over the full study site. While this might have introduced minor inaccuracies it removed clutter from the resulting maps and yielded a more complete picture.

We estimate that 207 Tg of organic carbon are stored in the upper 2 cm of seabed sediments over a mapped area of

1,129,952 km$^2$. Organic carbon stocks varied from 0.03 kg m$^{-2}$ to 0.45 kg m$^{-2}$ (Figure 3a). Stocks are lowest on the North Sea

shelf, the mid-Norwegian shelf, along the shelf edge and slope and parts of the southern Barents Sea shelf. Conversely, stocks are highest off the coasts of the Svalbard archipelago, on the Spitsbergen Bank and on the Central Bank.

## 3.2 Spatial patterns of fishing disturbance

Swept volume ratios (SVR), as an indicator of fishing intensity, ranged from 0 yr$^{-1}$ to 22.4 yr$^{-1}$ in the surface layer (Figure 3b), while the mean annual SVR was $(0.14 \pm 0.6)$ yr$^{-1}$. Values were highest along the southern rim of the Norwegian Trough and the innermost part of the Skagerrak. Elevated values of SVR were also associated with banks in the Norwegian and Barents Seas, while large parts of the Barents Sea, Norwegian Sea and Norwegian Trough experienced low to no fishing impact.

## 3.3 Annual cumulative disturbance of organic carbon stocks

The annual cumulative disturbance of organic carbon in the surface layer across the study region was estimated as 20.7 Tg yr$^{-1}$. Disturbance of organic carbon stocks varied spatially with organic carbon stocks and fishing disturbance (Figure 3c). Organic carbon disturbance ranged from 0 kg m$^{-2}$ yr$^{-1}$ where there is no fishing disturbance to 4.76 kg m$^{-2}$ yr$^{-1}$.

## 3.4 Meta-analysis results

The meta-analysis revealed statistically significant $(p < 0.0001)$ acute effects of mobile bottom fishing on organic carbon content in surface sediments (Table 1). Organic carbon content was reduced by 21.7% [95% CI = 15.9% to 27.9%] when compared with non-fished control areas or before bottom fishing took place. Impacts of bottom fishing on dry bulk density were not significant $(p = 0.93)$ and showed increases as well as decreases. Despite this, the impact on organic carbon densities was statistically significant $(p = 0.039)$ with decreases of 11.5% [95% CI = 0.6% to 21.2%].

## 3.5 Vulnerable organic carbon: How much and where

Unfished areas accounted for approximately 765,600 km$^2$, containing 139.2 Tg of organic carbon in the surface layer (Figure 3c). Of this, 18.7 Tg (1.9 – 33.5 Tg) were potentially vulnerable to mobile bottom fishing. Reductions in organic carbon density accounted for 16.0 Tg (0.8 – 29.5 Tg) of the vulnerable organic carbon. However, the choice of the value for the fraction of the sediment that resettles influenced all three impact scenarios (Table S1). In the low impact scenario, the estimated stocks of vulnerable organic carbon differed by a factor of 11.5 between the extremes of $P_{crd} = 0$ (no sediment resettles) and $P_{crd} = 1$ (all sediment resettles). In the mean and high impact scenarios, the estimated stocks varied by a factor of $\approx 2$.

Most of the vulnerable organic carbon is located in the Barents Sea off the coasts of the Svalbard archipelago and in the vicinity of the Spitsbergen and Central Banks (Figure 4). In the low impact scenario, there are also sizable amounts of vulnerable organic carbon to be found in the Skagerrak, since changes in sediment thickness have a relatively higher importance in these mud-dominated environments when changes relating to organic carbon density are relatively low. Overall, spatial patterns look similar regardless of fishing impact scenario while the magnitude of vulnerable organic carbon varies between scenarios.

Approximately 6.8 Tg (0.7 – 12.2 Tg) of vulnerable organic carbon are located in existing area-based protection measures (Figure 4). This equates to approximately 36% of the vulnerable organic carbon currently protected against the impacts of mobile bottom fishing.

Hotspots of vulnerable organic carbon were exclusively limited to the Barents Sea in the mean and high impact scenario, while the Skagerrak was a hotspot area in the low impact scenario (Figure 5). Hotspot areas increased from the low to the high impact
scenario from 64,928 km$^2$ to 81,552 km$^2$. Identified hotspots were similar between the mean and high impact scenario, while those identified in the low impact scenario differed to some extent. A limited number of coldspots in the Norwegian and southern Barents Seas was also found in the low impact scenario. The area of hotspots that is currently located within area-based protection measures amounts to 26,736 km$^2$ (4,160 – 30,208 km$^2$), equating to 34.5% (6.4 – 37.0%) of the total area.

**4 Discussion**

Approximately 207 Tg of organic carbon are stored in the surface sediments of the Norwegian continental margin. In areas where mobile bottom fishing activity has been detected, we find an annual cumulative disturbance of organic carbon stocks of 20.7 Tg yr$^{-1}$. Unfished areas cover 765,600 km$^2$ and contain between 1.9 and 33.5 Tg of vulnerable organic carbon (carbon that might be lost due to a fishing event in a currently unfished location) in the surface layer, based on a 0.6% to 21.2% reduction of organic carbon densities derived by a meta-analysis of published studies and reductions in sediment thickness due
to fishing-induced erosion. Hotspots of vulnerable organic carbon are found in the Barents Sea, off the coasts of the Svalbard archipelago and in the vicinity of Spitsbergen and Central Banks.

**4.1 Impacts of mobile bottom fishing on organic carbon stocks**

Impacts of mobile bottom fishing on seabed biogeochemistry have been investigated for more than three decades; however, a first global meta-analysis was only published recently (Tiano et al., 2024). The most frequently analysed parameter was organic
carbon content but there is a scarcity of studies investigating sedimentological parameters and neither porosity nor dry bulk density have been reported by Tiano et al. (2024). It follows from Eq. 1 that both parameters are of equal importance when considering impacts on organic carbon stocks. We have therefore conducted a new meta-analysis using studies that measured both organic carbon content and bulk density within the same study based on data in DISOM (Paradis et al., 2024) and a systematic review search (Felgate et al., 2024). We have found statistically significant relative reductions of organic carbon
densities (-11.5%) in surface sediments although less pronounced than in organic carbon content (-21.7%).

This is important in two ways: Firstly, it means that mobile bottom fishing is reducing organic carbon stocks, something that has not been shown before in fishing impact studies based on field experiments or the meta-analysis by Tiano et al. (2024). Secondly, it means that measuring changes in organic carbon content alone is not sufficient when estimating the impact of mobile bottom fishing on organic carbon stocks and might lead to incorrect results. Based on the data from our meta-analysis
(Table 1), using impacts on organic carbon content instead of organic carbon density would lead to estimates that are too high

by a factor of almost 2. To make things worse, organic carbon content (measured in weight-% or equivalent) is frequently referred to as organic carbon concentration, which denotes the mass of organic carbon per volume of sediment and is measured in g C cm$^{-3}$ or equivalent (Flemming and Delafontaine, 2000) and is the same as organic carbon density (Eq. 3). While changes in organic carbon concentration (density) directly relate to changes in organic carbon stocks, this is not the case for changes in

organic carbon content. Using the terms organic carbon content and concentration interchangeably blurs this important difference. We therefore call on researchers to use consistent terminology and always report the units associated with the parameters they are reporting.

In addition, changes to the sediment thickness due to erosion of the top layer of seabed sediment by mobile bottom fishing need to be considered when investigating fishing impacts on organic carbon stocks (eq. 8). Assuming $P_{crd} = 0.87$ (Sala et al.,

2021). Our results indicate that, with the notable exception of the low impact scenario, the estimated reductions in organic carbon stocks are largely due to reductions in organic carbon density, while thickness changes only play a minor role. However, changes to the sediment thickness are sensitive to the choice of $P_{crd}$, which is poorly constrained.

## 4.2 Management implications

In the context of nature-based solutions to climate change there are two types of measures that are discussed: avoided emissions

and carbon removal. The latter is important as it could potentially contribute to negative emissions (carbon dioxide removal), which are required to neutralize any residual $CO_2$ emissions on the path to net zero emissions. Avoided emissions should only be considered as a climate mitigation measure when they are additional to a business-as-usual baseline, e.g., by focussing on ecosystems at a high risk of losing carbon stores (Cook-Patton et al., 2021). Protecting seabed areas with high organic carbon stocks from disturbance caused by mobile bottom fishing has been proposed as one such measure (Jankowska et al., 2022). It

has been suggested that protection measures should be generally preferred over management (e.g., temporal closures or restrictions on fishing gear) and restoration due to high per-area mitigation that can be realised quickly, at a relative low cost and with many co-benefits (Cook-Patton et al., 2021). Easing the pressure on important seabed organic carbon stocks by implementing carbon protection zones is therefore the most prominent management option currently discussed (Epstein and Roberts, 2022; Porz et al., 2024; Sala et al., 2021).

In areas that are currently fished, we find that the annual cumulative disturbance of organic carbon stocks in Norway (20.7 Tg yr$^{-1}$) is considerably lower than in the United Kingdom (109 Tg yr$^{-1}$). This difference might be in part due to the fact that we only considered the surface layer. However, it is likely that the difference can also be attributed to the overall higher fishing intensity in the waters around the United Kingdom (mean SVR = 0.3 yr$^{-1}$; Epstein and Roberts (2022)) as compared to Norway (mean SVR = 0.14 yr$^{-1}$). Priority areas for managing the impact of mobile bottom fishing on seabed organic carbon stocks

could be identified following the methodology outlined by Epstein and Roberts (2022). However, the gain in potentially avoided emissions is likely to be much smaller in Norway than in the United Kingdom due to the lower levels of cumulative disturbance of organic carbon stocks. Additionally, such spatial protection measures could displace fishing fleets, and without appropriate reductions in total allowable catch this could partially offset or negate the positive effects of local seabed protection

(Greenstreet et al., 2009). Protecting currently unfished seabed areas high in organic carbon and at risk of being impacted might therefore be an interesting option that should be explored as well.

Our study on vulnerable organic carbon in currently unfished areas is adapted from the concept of irrecoverable carbon in Earth's ecosystems (Goldstein et al., 2020; Noon et al., 2022). These authors propose three dimensions of ecosystem carbon stocks: manageability at the local scale, magnitude of vulnerable carbon, and recoverability of ecosystem carbon if lost. Seabed organic carbon stores are in general manageable through fisheries management (Andersen et al., 2024). We have proposed a definition of vulnerable organic carbon and applied it to currently unfished areas of the Norwegian margin. Our analysis shows that between 1.8 and 29.6 Tg of organic carbon stored in surface sediments are potentially vulnerable to acute fishing impacts. Hotspots of vulnerable organic carbon are exclusively located in the Barents Sea in all three scenarios. The Barents Sea is one of the polar regions where climate and ecosystem change is most pronounced (Gerland et al., 2023), undergoing temperature increase and receding seasonal sea ice cover (Ingvaldsen et al., 2021). Receding sea ice has opened up pristine areas for fishing, which has expanded into the northern Barents Sea, based on a comparison of the results from a model hindcast (1982 - 2011) with recent fishing activity (2013 – 2018) (Fauchald et al., 2021). Vulnerable organic carbon stored in seabed sediments in the northern Barents Sea could be at high risk of being lost due to expansions of mobile bottom fishing if left unprotected. Between 19% and 46% of the hotspot area are currently located within area-based protection measures, noting that these areas had the objective to protect biodiversity rather than vulnerable organic carbon. Additional carbon protection zones could be focussed on hotspots of vulnerable organic carbon which are currently located outside the existing area-based protection measures.

Poleward shifts of fishing activity due to diminishing sea ice cover have also been hindcasted for the Bering Sea and the Sea of Okhotsk (Fauchald et al., 2021). We expect that our methodology is also applicable to those shelf seas, and it should therefore be possible to identify vulnerable organic carbon stocks in these areas. In addition, increased fishing activity due to projected future sea ice loss might be expected in the Chukchi Sea, the Canadian Archipelago, Hudson Bay, the coastal areas of Newfoundland and Labrador, the Greenland coast and the Kara Sea (Cruz et al., 2024; Fauchald et al., 2021). Such information could be critical to protecting vulnerable organic carbon in a rapidly changing Arctic. However, methods to derive predictive maps of (vulnerable) organic carbon in the future would need to factor in additional drivers of organic carbon gain and loss, such as changes in surface primary productivity, bottom water temperature and oxygen among others.

### 4.3 Method appraisal and limitations

Our study focusses on currently unimpacted organic carbon stocks and is hence complementary to previous work that has aimed to map and quantify fishing impacts in currently fished areas. Protecting seabed sediments with large organic carbon stocks from disturbance by mobile bottom fishing might be considered as a potential natural climate solution to avoid emissions (Jankowska et al., 2022). However, it is conceivable that such protections could lead to a redistribution of fishing effort that might impact currently undisturbed stocks, thereby limiting or even negating the benefits of protection. Moreover, truly pristine organic carbon stocks might become vulnerable to fishing impacts where sea-ice is retreating due to Arctic warming (Fauchald et al., 2021). Protecting these pristine stocks should have high priority, as protection is a more efficient strategy for mitigating

climate change than restoration (Ascenzi et al., 2025; Cook-Patton et al., 2021). We therefore believe that it is important to also focus on these currently unimpacted or even pristine organic carbon stocks. While the presented methodology provides a framework for quantifying and mapping vulnerable organic carbon in unfished areas, we acknowledge that there are limitations, which we will discuss in the following.

The fishing intensity data obtained from ICES (2021) is limited to fishing vessels above 12 m length. Smaller vessels mainly operating in inshore waters are not captured. Since our study area does not include inshore areas, but is limited to offshore areas seaward of the baseline, we do not expect our results to be biased because of the lack of VMS data for vessels under 12 m. However, vessel monitoring system data from Portugal, Iceland and Norway are not included in the ICES spatial layers (as the submitted data did not pass the quality check), hence the distribution and intensity of fishing effort may be underestimated. We draw caution to the fact that the actual fishing impact might be higher than captured in the fishing intensity data we used. Data outputs from ICES (2021) assume a uniform distribution of mobile bottom fishing within each grid cell. This assumption will apply when mobile bottom fishing is evaluated over longer periods, since the random distribution on the annual time scale will become a uniform distribution at a time scale of multiple years (Ellis et al., 2014). Since we use 12 years of fishing data, we do assume that the assumption holds.

The meta-analysis is currently based on nine studies located in the Baltic Sea, the Mediterranean and the Gulf of Maine. These study sites differ from the Norwegian margin to some degree with regard to environmental conditions (Figure S1), but we believe that these differences are not large enough to invalidate our study results. Nevertheless, there clearly is a need for more studies from different environments globally that measure organic carbon content together with dry bulk density or porosity to calculate organic carbon density or measure organic carbon densities directly. In the meantime, we use the 95% confidence interval limits to estimate the likely range of the impacts.

Our analysis is limited to upper 2 cm of the sediment column, as statistically significant impacts of mobile bottom fishing on biogeochemistry and sedimentology have been primarily found in this surface layer (Tiano et al., 2024). However, the average penetration depth of otter (2.44 cm) and beam trawls (2.72 cm) is slightly larger (Hiddink et al., 2017), and some gear components reach much deeper (Eigaard et al., 2016). Our results might therefore be seen as a conservative estimate until more studies are available that can be used in meta-analysis studies to detect impacts more reliably in deeper layers.

Our analysis does account for the local bulk response of organic carbon stocks to fishing disturbance. As such, it cannot differentiate between the different processes as presented in the introduction including transport and redeposition of organic carbon in nearby locations (Porz et al., 2024; Zhang et al., 2024). Disentangling these processes requires 3D-coupled numerical models as used by Porz et al. (2024) and Zhang et al. (2024). However, such models are resource and time demanding.

Estimates of changes in the sediment thickness due to mobile bottom fishing are sensitive to the proportion of the sediment that resettles after resuspension ($P_{crd}$). This parameter is, however, poorly constrained and it is unlikely that it is a globally applicable constant. Our methodology could be further improved by modelling this parameter over the area of interest and treating it as a variable rather than a constant in eq. 9.

## 5 Conclusion

Our study focuses on currently unimpacted organic carbon stocks and we develop a methodology to estimate the amount of vulnerable organic carbon in these areas. We argue that protection of vulnerable organic carbon within currently unfished areas that are at high risk of being lost if fished should have a high priority. Pristine organic carbon stocks might be particularly at risk in areas where the footprint of mobile bottom fishing extends into increasingly deeper waters on continental slopes (Watson and Morato, 2013) and where rapid Arctic warming leads to the retreat in sea ice cover (Fauchald et al., 2021). Protection of vulnerable organic carbon in currently unfished areas also ensures that fishing displacement caused by e.g. protection measures will not impact vulnerable stocks.

**Code availability**

All original code has been deposited at Github and is publicly available at https://github.com/diesing-ngu/fishing_impacts_on_carbon as of the date of publication.

**Data availability**

Output data have been deposited at Zenodo and are publicly available as of the date of publication at https://zenodo.org/records/14614698.

**Supplementary material**

Document S1: Figure S1 and Table S1

**Author contribution**

Conceptualization, M.D.; methodology, M.D., and M.S.; software, M.D.; validation, M.D.; investigation, M.D., and M.S.; data curation, M.D.; writing—original draft, M.D.; writing—review & editing, M.D., M.S., T.T., L.R.B., and Ø.G.M.; funding acquisition, T.T. and L.R.B.

**Competing interests**

The authors declare that they have no conflict of interest.

## Acknowledgements

This work was supported by the Norwegian seabed mapping programme Mareano.

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

**Table**

**Table 1: Acute effects of mobile bottom fishing on sediment parameters in the surface layer (0 – 2 cm). lnRR is the log response ratio, SE the standard error, CI stands for confidence interval and n denotes the number of independent studies.**

| *Variable* | lnRR | SE | p-value | 95% CI | n |
|---|---|---|---|---|---|
| *Organic carbon content* | -0.245 | 0.042 | <0.0001 | -0.327 to -0.162 | 9 |
| *Dry bulk density* | -0.015 | 0.167 | 0.93 | -0.341 to 0.312 | 9 |
| *Organic carbon density* | -0.1222 | 0.0593 | 0.039 | -0.238 to -0.006 | 9 |

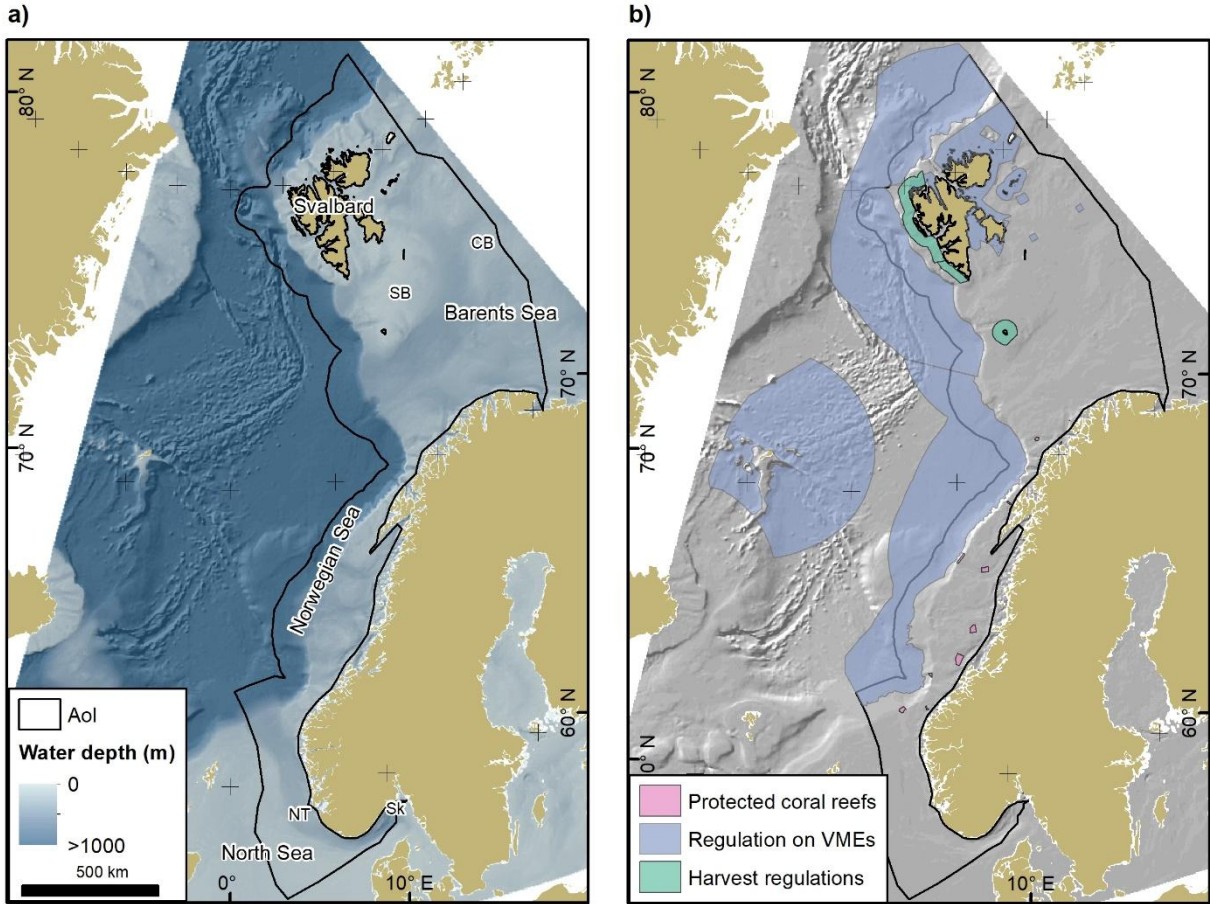

**Figure 1: a) Overview of the area of interest (AoI). Water depths (GEBCO Bathymetric Compilation Group, 2019), regional seas and locations mentioned in the text are indicated. CB – Central Bank; NT – Norwegian Trough; SB – Spitsbergen Bank; Sk – Skagerrak. b) Area-based protection measures based on the regulation on the protection of coral reefs, §58 of the harvest regulations, and the regulation on vulnerable marine ecosystems (VMEs). See methods for additional information on area-based protection measures. Source of land areas: ESRI.**

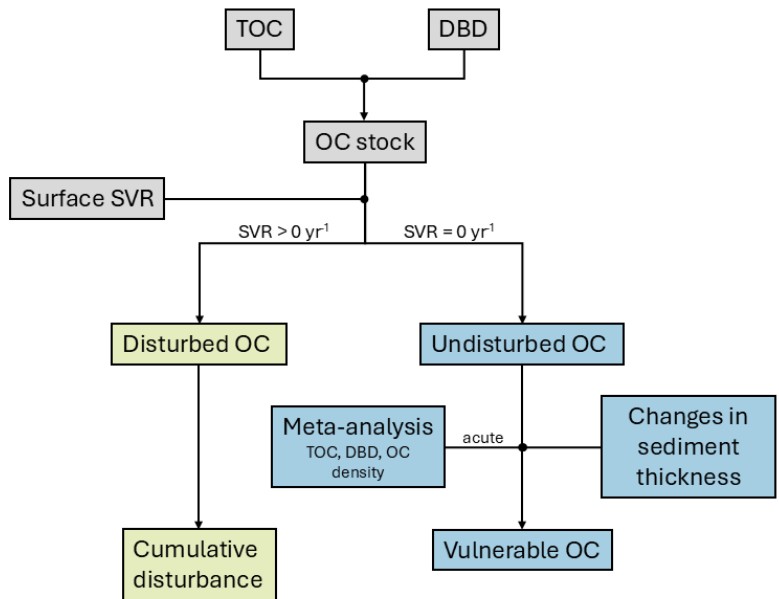

**Figure 2: Flow diagram outlining the workflow of the current study. TOC – total organic carbon content, DBD – dry bulk density, OC – organic carbon, SVR – swept volume ratio.**

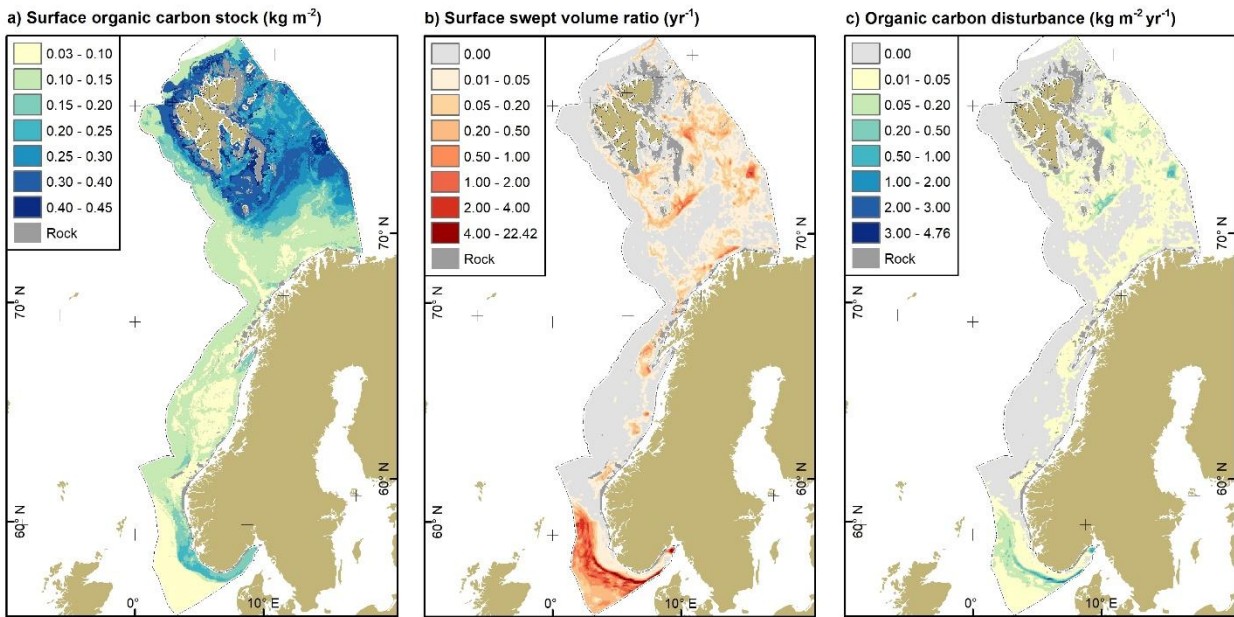

**Figure 3: a) Surface organic carbon stocks estimated from predicted organic carbon content and dry bulk density. b) Surface swept volume ratio of beam trawls, demersal seines and otter trawls as a mean of the years 2009 to 2020. c) Annual cumulative disturbance of organic carbon stocks based on a) and b). Rock refers to hard substrates, mainly rock and boulders. Source of land areas: ESRI.**

650

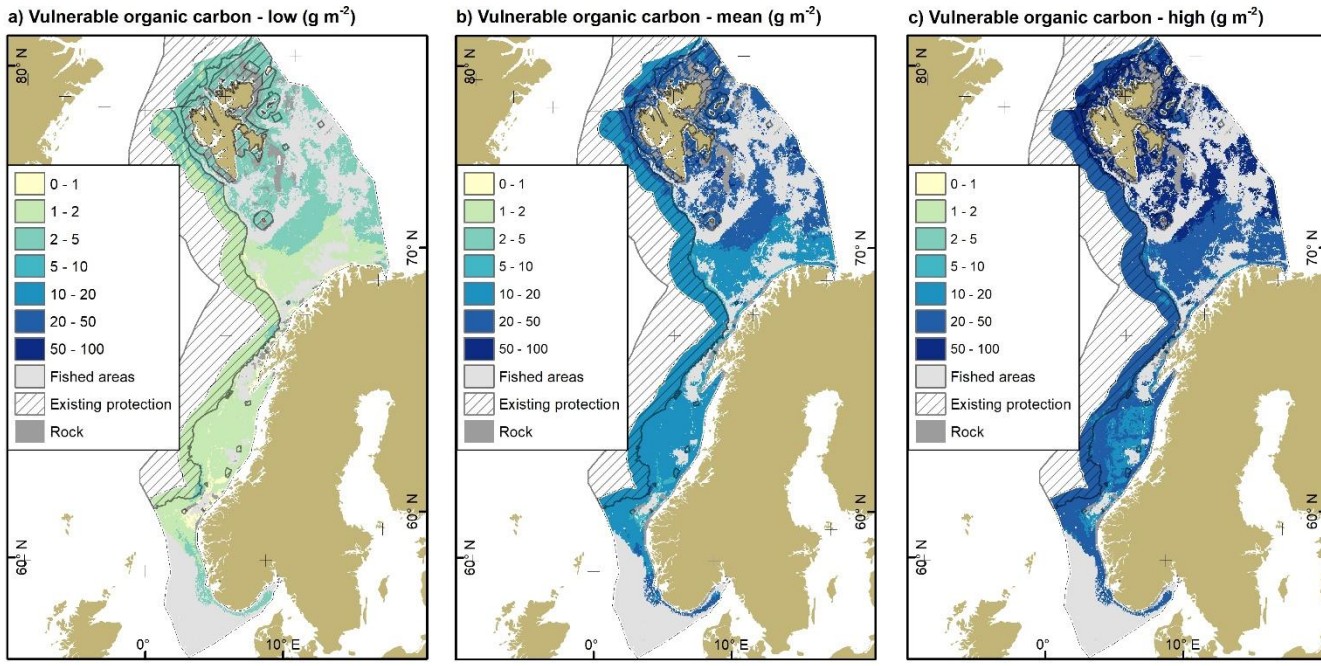

**Figure 4: Vulnerable organic carbon stocks of the surface layer for the three fishing impact scenarios: a) low, b) mean and c) high. Existing area-based protection measures as in Figure 1b and described in Methods. Rock refers to hard substrates, mainly rock and boulders. Source of land areas: ESRI.**

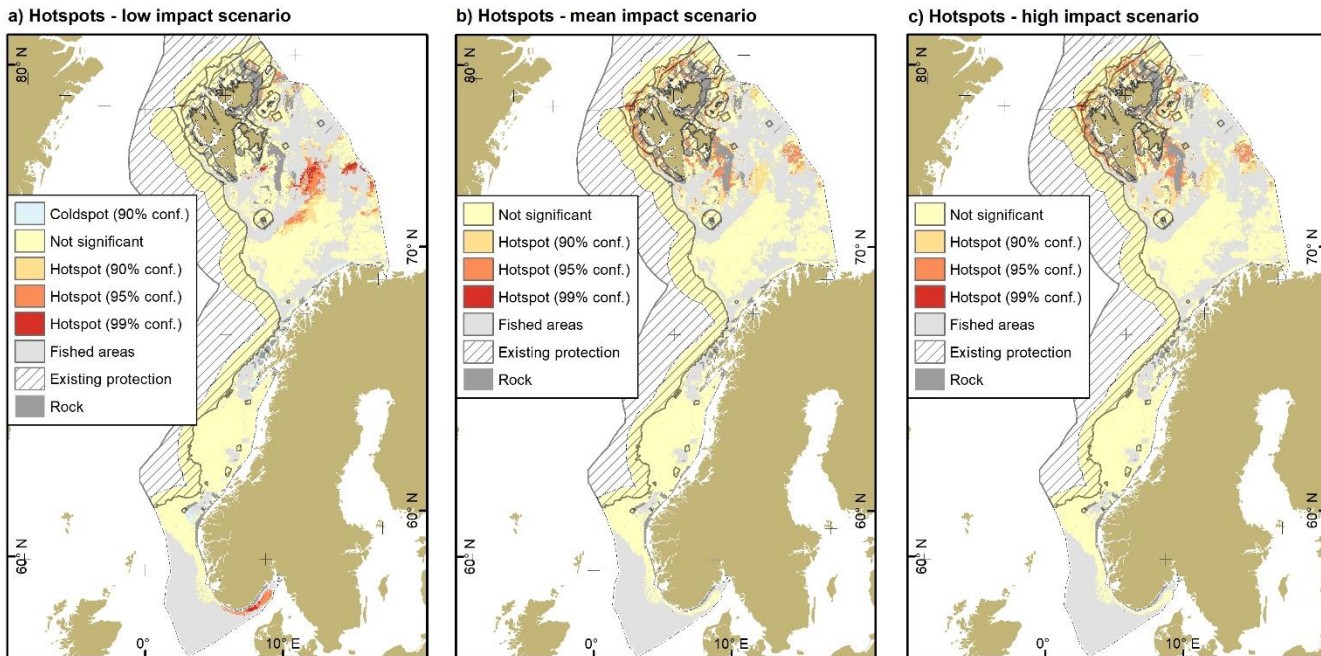

Figure 5: Hotspots of vulnerable organic carbon for the three impact scenarios: a) low, b) mean and c) high. Coldspots of vulnerable organic carbon were present only for the low impact scenario along the northern coast of mainland Norway. Existing area-based protection measures as in Figure 1b and described in Methods. Rock refers to hard substrates, mainly rock and boulders. Source of land areas: ESRI.