# Peer review of "Mapping organic carbon vulnerable to mobile bottom fishing in currently unfished areas of the Norwegian continental margin"

_EGUsphere, 2025_

## Author Response (AR1)

**Referee #1:**

I have reviewed an earlier version of this manuscript once before for another journal. I can see that the authors have addressed a few of my original concerns in this new submission, but not others. Therefore, I have repeated my remaining comments below, along with a few new ones.

In general, I find this to be an interesting and relevant study that adds to the ongoing discussion on seabed carbon management, and I believe it will be useful for many researchers and policymakers interested in this topic. I am not an expert on the machine learning method used in the mapping, but I tend to believe it is trustworthy since it is based on previously published studies. Nevertheless, I do have one general criticism related to the impact calculation (see general comment #1) and a few issues with the interpretation of the results in relation to management recommendations (see general comments #2+#3). I hope that the authors can alleviate my concerns by expanding/adapting the manuscript.

General comments:

The authors estimate bottom fishing impacts on carbon based on (1) assumptions about resuspension and settling and (2) measured changes in carbon density, and they treat both as separate, independent effects. In reality, however, the two are intrinsically linked: Sediment disturbance partly oxygenates organic carbon during resuspension and mixing, leading to enhanced remineralization, and partly winnows fine particles following resuspension. Both can lead to a decreased carbon content. In other words, the effects of sediment resuspension may already be contained in the measured carbon losses (i.e., Eq. (5) is not quite correct, because $\delta\rho_{oc}$ is a function of $\delta d$). I suspect that treating carbon loss and resuspension separately may cause an overestimation of the net effect.

**Reply: We understand that it might initially appear as if we are "double counting" by treating the impacts on organic carbon density and sediment thickness separately. However, we can assure referee #1 that this is not the case. The starting point of our considerations is the equation to calculate organic carbon stocks (eq. 1), which involves organic carbon content, dry bulk density and the sediment thickness. The product of organic carbon content and dry bulk density is the organic carbon density. We estimate changes to this metric by applying a meta-analysis and find a statistically significant reduction of 11.5% on average. Using this value alone would, however, be insufficient, as organic carbon density specifies the mass of organic carbon per volume of sediment but not the changes to the volume under consideration, since fishing also leads to erosion of sediment and hence a reduction of sediment thickness. The changes in sediment thickness are thus a scaling factor that accounts for this erosion, as stated in line 229.**

In summary, we are not treating carbon loss and resuspension separately in the way described by the referee. Rather, we account for the effects of resuspension on organic carbon density and the sediment thickness separately.

Action: We have added more information in chapter 2.8 to explain this: "Besides changes in organic carbon density, mobile bottom fishing also leads to erosion of sediment and hence a reduction of the sediment volume under consideration. Changes in sediment thickness act as a scaling factor that accounts for this erosion of the seabed."

Though the effects are difficult to separate without dedicated process-based modelling, I recommend the authors discuss this and associated uncertainties by showing both effects separately, i.e., what the effects of only resuspension vs. only carbon loss are.

Reply: As stated in chapter 4.3 on limitations, our method cannot differentiate between the different processes as presented in the introduction including transport and redeposition of organic carbon in nearby locations (Porz et al., 2024; Zhang et al., 2024). Disentangling these processes requires 3D-coupled numerical models as used by Porz et al. (2024) and Zhang et al. (2024); however, these are very time and resource-demanding and require specialist skills. Our approach might be easier and quicker to implement elsewhere.

Action: Our methodology cannot disentangle the mentioned processes. However, we now show the effects of mobile bottom fishing on organic carbon density and sediment thickness separately. This can be achieved by setting $P_{crd} = 0$ in eq. 9, i.e. all resuspended sediment resettles.

When making these changes to the R script, we discovered two mistakes. The first one relates to the conversion of units when estimating the mass of suspended sediment after Oberle et al. (2016). The equations in that publication require data on silt-clay content in weight-%, while the data we used was expressed as a dimensionless fraction. The second mistake derives from inconsistencies in terminology. In the meta-analysis the lower bound of the confidence interval (Table 1) relates to the high impact scenario (due to the negative sign of the log response ratio), while the lower linear limit of resuspended sediment after Oberle et al. (2016) relates to the low impact scenario. These mistakes have now been resolved and the changes impact estimates of vulnerable organic carbon and associated hotspots (chapter 3.5, Figures 4 and 5). However, the changes are relatively small (e.g. 18.7 Tg instead of 16.0 Tg of vulnerable organic carbon in the mean impact scenario) and do not affect the overall interpretation and significance of the results. We did notice, though, that the estimation of change in sediment thickness is now sensitive to the choice of the parameter $P_{crd}$ and discuss this in chapter 4.3.

The authors base much of their discussion and recommendations around the estimated "recovery time", i.e. the time until the prior sediment carbon stock has been reached following one bottom fishing event (SVR=1), but I question whether this is really a meaningful metric for spatial management. In my view, when it comes to seabed carbon management, recovery of accumulation rate is more relevant, that is, how quickly (if at all) the seafloor can regain its function as a carbon sink following protection. Perhaps the authors can clarify this issue in their discussion by distinguishing these separate aspects of (1) standing stock depletion and (2) sequestration rate reduction, in relation to potential climate and ecosystem impacts (see also my detailed comment #5).

**Reply: We do not think it is entirely true to say that much of the discussion and recommendations are based on the estimated recovery time. This is in fact only true for the last paragraph of chapter 4.2 (lines 389 – 405). Most of the discussion and recommendations are instead informed by the estimates of vulnerable organic carbon and the associated hotspot analysis.**

**The methodology we used was adapted from the concept of irrecoverable carbon (Goldstein et al., 2020), which was applied to terrestrial ecosystems. Irrecoverable carbon refers to the stores of carbon in nature that are vulnerable to release from human activity and, if lost, could not be restored by 2050 (when the world must reach net-zero emissions). It can be estimated by subtracting carbon that recovers within a certain timeframe from vulnerable carbon. However, we have refrained from estimating irrecoverable carbon, as organic carbon stored in seabed sediments is not in direct contact with the atmosphere and the climate impacts are less certain then in the case of terrestrial ecosystems. Nevertheless, we found that some areas would not recover at all (or only very slowly) and thought that this was worth highlighting. Since the results from the recovery analysis are not central to our discussion, we could remove this part of the analysis if referee #1 maintains that these results have limited relevance.**

**Action: We have removed all sections relating to recovery time.**

I would like to see a discussion of "loss" due to (1) resuspension and transport, which does not necessarily affect the seabed's overall capacity for carbon storage/sequestration and is therefore not as relevant for climate or ocean acidification, vs. (2) loss due to remineralization, which is more relevant and is what most previous studies on the topic were concerned with. Right now, this is only mentioned in the introduction, and while I understand that estimating remineralization following disturbance might be beyond the scope of this study, I do recommend that the authors at least address the issue in their discussion in relation to their own results.

**Reply: This has been addressed in the last paragraph of chapter 4.3 on the limitations of our approach (lines 440 – 444): "Our analysis does account for the local bulk response of organic carbon stocks to fishing disturbance. As such, it cannot differentiate between the different processes as presented in the introduction including transport and redeposition of organic carbon in nearby locations (Porz et al., 2024; Zhang et al., 2024). Disentangling these processes requires 3D-coupled numerical models as used by Porz et al. (2024) and Zhang et al. (2024). However, such models are resource and time demanding." We also define loss as "the reduction of local organic carbon stocks" in the last paragraph of the introduction (line 89).**

**Action: As mentioned above, we are now showing the effects due to changes of organic carbon density and sediment thickness separately. We have also added additional text to chapter 4.1 discussing the relative effects of these two parameters and their relevance for future studies.**

Detailed comments:

191: consider making this a numbered equation.

**Reply: Could be, if required. We are impartial to that issue.**

**Action: So far, no action has been taken.**

289-290: As I understood from 2.7, carbon densities were calculated from OC content and DBD according to Eq. (3). Doesn't the impact on OC densities then simply follow from the decrease in carbon content in this case, with no significant influence from bulk density? The way it is presented now reads as though the impact on OC density has been confirmed independently. The same is repeated in 338-339. While it makes sense to focus on OC density for gauging overall impacts, I find the way the meta-analysis is presented now to be misleading and I recommend removing the statistical analysis for impacts on OC density.

**Reply: We did not calculate the effect on organic carbon density by multiplying the effects on organic carbon content and dry bulk density. Organic carbon densities were calculated according to eq. 3 prior to the meta-analysis. We have then carried out meta-analyses on all three metrics. What we found were statistically significant reductions in organic carbon content and organic carbon density, but not in dry bulk density. Just because the impacts on dry bulk density were not statistically significant does not mean that there weren't any. Decreases in organic carbon content must not be equated to decreases in organic carbon density. This is what has, implicitly or explicitly, happened in previous studies but one takeaway message of our manuscript is that this is not correct (see also the reply to the subsequent comment).**

**Action: The estimated changes in OC density due to fishing derived through the meta-analysis are a fundamental requirement to carry out the analysis. Following the recommendation of removing this analysis is not feasible. Therefore, no action was taken.**

340-349: The distinction between density and concentration is important and I agree with the authors' call for consistency in terminology.

**Reply: This is not what we state. We use definitions from Flemming and Delafontaine (2000): The term content denotes a mass per unit mass (g/kg or similar). The term concentration denotes a mass per unit volume (g/cm³ or similar). From this it follows that organic carbon content and organic carbon density (measured in g/cm³) are different entities, while organic carbon concentration and organic carbon density are the same. Unfortunately, organic carbon content and concentration have been used interchangeably in previous studies (including one of the first author). We wanted to highlight this confusion and the need for consistent terminology and believe that the definitions of Flemming and Delafontaine (2000) are useful in this context.**

I also agree that bulk density should be reported. However, I disagree with the framing in l.340-343, which implies that previous assessments of bottom fishing impacts may be erroneous for not distinguishing density and concentration, and that this study is the first to show depletion of carbon stocks by bottom fishing. As far as I am aware, previous studies have accounted for porosity/bulk density when calculating stocks (e.g. Zhang et al., 2024, Nat. Geosci.) Though changes to bulk density by bottom fishing have not been accounted for in previous studies, bottom fishing does not seem to change the bulk density of the sediment according to the authors' own results (Table 1), meaning that change to carbon concentration is indeed the driving factor behind carbon stock depletion by bottom fishing. Therefore, I don't think the authors have really shown that impacts on bulk density need to be accounted for when gauging bottom fishing impacts on carbon stocks. Their results rather imply the opposite (which would also be a useful result). In any case, I feel this section should be toned down or clarified.

**Reply: We stand by our claims that (1) this fishing impact study is the first to show that mobile bottom fishing is reducing organic carbon stocks and that (2) measuring changes in organic carbon content alone is not sufficient when estimating the impact of mobile bottom fishing on organic carbon stocks and might lead to incorrect results.**

**Firstly, the comprehensive global meta-analysis published last year by Tiano et al. does not report organic carbon stock or density as a variable. We are not aware of any newer studies which might have reported these variables and hence this is the**

first study to do so. The confusion might arise from what is meant by fishing impact studies. We refer here to studies based on field experiments (with before-after (BA), control-impact (CI) or BACI design), comparative CI studies and meta-analyses thereof. We do not include modelling studies. We could define this more clearly in the text.

**Action: We have updated the text. It now reads: "[...] mobile bottom fishing is reducing organic carbon stocks, something that has not been shown before in fishing impact studies based on field experiments or meta-analyses."**

**Secondly, changes in organic carbon content (not concentration) alone are not sufficient to estimate changes in organic carbon stocks. The claim that "bottom fishing does not seem to change the bulk density of the sediment according to the authors' own results (Table 1)" is incorrect. As mentioned before, we only state that "impacts of bottom fishing on dry bulk density were not significant (p = 0.93) and showed increases as well as decreases" (lines 288 – 289). This is not the same as no changes to dry bulk density. Hence, dry bulk density and organic carbon content should be used to calculate organic carbon density, which could in turn be used in a meta-analysis.**

**To illustrate the effect of using changes in organic carbon content instead of organic carbon density: We found statistically significant reductions of 21.7% and 11.5%, respectively (chapter 3.4). Using the former instead of the latter would lead to reductions that are too high by almost a factor of 2. We could add this example to the text to make clearer why we think it is not sufficient to measure changes in organic carbon content alone.**

**Action: A sentence has been added to section 4.1 reading: "Based on the data from our meta-analysis (Table 1), using impacts on organic carbon content instead of organic carbon density would lead to estimates that are too high by a factor of almost 2."**

**See also the comment by referee #2 on the same matter: "That's a good point that will be important for other scientists to account for in the future".**

362-366: It may be useful to consider the total area considered for UK and Norway here and calculate OC disturbance/area to make a more meaningful comparison.

**Reply: We agree that including information about the total fished area would be useful for the comparison and will add this to the revised manuscript.**

**Action: Unfortunately, the relevant information was not available from Epstein and Roberts (2022). Therefore, no action was taken.**

397-405: As per my general comment, I question the utility of the "recovery"-concept when it comes to seabed carbon management. In my understanding, the difference

between "recoverable" and "irrecoverable" seabed carbon is that accumulation rates are much lower for irrecoverable stocks. However, carbon lost/remineralized from a "recoverable" site will have the same potential climate impact as carbon lost from an "irrecoverable" site. The net flux will not change just because one site has active accumulation and the other does not; net loss over time is the same in both cases. The distinction would only be important if future carbon accumulation rates also changed as a consequence of past bottom fishing, e.g. due to changes to benthic community structures and associated ecosystem functions, or if future carbon accumulation was somehow otherwise limited, e.g. by accommodation space (unlikely considering sea level rise).

**Reply: We clearly define the term irrecoverable carbon in the text (lines 397–400). It refers to carbon stocks in nature that are vulnerable to human-driven release and, if lost, could not be restored by 2050—the target year for achieving global net-zero emissions. Our analysis shows that some areas of the seabed either do not recover at all or recover only very slowly, which we considered important to highlight. While the net long-term carbon loss is the same for both recoverable and irrecoverable seabed carbon, the distinction lies in ecosystem resilience to trawling. Where carbon accumulation rates exceed resuspension from trawling, and trawling does not substantially reduce accumulation, recovery is possible meaning these recoverable areas can help buffer the climate impact of trawling. It is therefore critical to distinguish the fraction of seabed that can recover from trawling disturbance from the fraction that cannot.**

**Action: As mentioned previously, all sections relating to recovery have been removed.**

4: the choice of different color mappings for the three scenarios makes it difficult to compare them among each other. The authors may consider choosing intervals that can be used for all three panels (perhaps using non-linear scaling/intervals).

**Reply: This could be done.**

**Action: Figure 4 has been altered so that all three panels have the same colour mappings.**

**Referee #2:**

**General comments**

Good paper on an important topic. I don't have much negative things to say about it.

I am slightly concerned though about the extrapolation from the meta-analysis so perhaps there could be some clarification.

**Reply: Please see chapter 4.3, lines 429 – 434: "The meta-analysis is currently based on nine studies located in the Baltic Sea, the Mediterranean and the Gulf of Maine. These study sites differ from the Norwegian margin to some degree with regard to environmental conditions (Figure S1), but we believe that these differences are not large enough to invalidate our study results. Nevertheless, there clearly is a need for more studies from different environments globally that measure organic carbon content together with dry bulk density or porosity to calculate organic carbon density or measure organic carbon densities directly. In the meantime, we use the 95% confidence interval limits to estimate the likely range of the impacts."**

**Action: None**

Many studies show a removal of OC after trawling but the amount of this which just lands somewhere else and is not fully degraded depends on many factors. Resuspended OC will undergo faster mineralization so while it is unlikely that all resuspended OC will just turn into CO2, the OC that gets redeposited will likely be degraded to some degree. I do imagine a net loss of OC after trawling but the amount of which that is redeposited is uncertain so it would be good to discuss this at least. For example, in Porz et al., 2024, they discuss how net amounts of carbon is ultimately (through mineralization) but their model on local scales shows increased carbon in several areas where you have re-deposition after resuspension. Perhaps there is something I missed that may be able to explain this but if not, please provide this caveat that the OC reductions are not necessarily net declines in OC but site specific declines.

**Reply: Please see chapter 4.3, lines 440 – 444: "Our analysis does account for the local bulk response of organic carbon stocks to fishing disturbance. As such, it cannot differentiate between the different processes as presented in the introduction including transport and redeposition of organic carbon in nearby locations (Porz et al., 2024; Zhang et al., 2024). Disentangling these processes requires 3D-coupled numerical models as used by Porz et al. (2024) and Zhang et al. (2024). However, such models are resource and time demanding. " This would preclude an analysis of trawling impacts on seabed carbon in regions unless they are data-rich and advanced numerical models are developed and parameterized with real data for specific regions.**

**Action: None**

Also, I wish other experts on this topic would discuss this more but I think that it is important to think about the potential long-term consequences of speeding up the carbon mineralization through trawling and how much that actually matters on climate relevant time-scales (for example, we often talk about needing 100+ years to consider

carbon sequestered long-term). If much of what is disturbed is still labile/reactive OC, then it might be re-mineralized naturally over the course of months and years without disturbance. Is this point relevant for discussion somewhere?

**Reply: Reactivity is indeed an important aspect to consider. However, we feel it is currently not universally defined what terms like labile, recalcitrant and refractory mean. There are definitions based on thermal reactivity (Smeaton and Austin, 2022), while modellers often use rate constants to define the different reactivity classes. It is, however, unclear how these relate to each other. The meta-analysis by Tiano et al. (2024) also included proxies for labile organic matter (chlorophyll-a, pigments etc.), but these components might indeed be very labile and be mineralised naturally over short time periods. For that reason, we did not include them in the analysis but instead considered the bulk effects of fishing on organic carbon stocks.**

**Action: From a practical point of view, there is also a lack of studies that have investigated fishing impacts on reactivity classes (e.g. as defined by Smeaton and Austin, 2022) that can be mapped spatially. We have added a sentence to the methods section describing the meta-analysis: "Given the available data, we restrict our study to fishing impacts on bulk organic carbon and make no attempt to account for different reactivity fractions."**

**Detailed comments**

**Abstract**

15 It's not clear what kind of analysis was conducted to get these estimates. Can you add some words to describe the modelling done to get to the 139.2 Tg of OC?

**Reply: We could add that we used modelling and spatial prediction techniques to estimate the organic carbon stock and fishing data from ICES to establish unfished areas. Based on both, we worked out the stock.**

**Action: The relevant sentence has been modified: "Using these data and spatial prediction methods, we estimate that the surface sediment layer (0 – 2 cm) in unfished areas covering 765,600 km$^2$ contains 139.2 Tg of organic carbon."**

Also, I would try to rephrase where the meta-analysis is described to show readers specifically that the meta-analysis estimated the decrease in OC to get to the 16.4 Tg value.

**Reply: It is a little more complicated. Based on the meta-analysis we estimated reductions of 11.5% on average. We also considered erosion of sediment due to bottom fishing, i.e. by how much the sediment volume would be reduced. Both**

were combined with the estimated stock in unfished areas to work out the vulnerable carbon. We can try to include this as well but remember this is an abstract with a word limit.

**Action: The relevant sentence has been modified: "Based on data from a meta-analysis of demersal fishing impacts on organic carbon density and estimated reductions in sediment thickness due to fishing-induced erosion, we estimate that 18.7 Tg (1.9 – 33.5 Tg) of organic carbon might be vulnerable to mobile bottom fishing in a scenario where each grid cell is fished evenly over the entire area and down to the full depth of the surface layer."**

This aspect isn't so clear and also consider that work from Black et al., 2022 and Smeaton et al., 2024 have a different definition of vulnerability (pressures x reactivity; it's more complicated but something like that). Upon the first read, I kept wondering: What makes OC vulnerable here exactly? Lability and proximity to the seabed surface? I think you just mean the amount of OC which can potentially be removed, no?

**Reply: Our definition of vulnerable carbon is based on the concept of irrecoverable carbon (Goldstein et al., 2020). The definition is given in chapter 2.8. We could try to include this information as well, but it might complicate and lengthen the abstract.**

**Action: It might be easiest in this case to replace "vulnerable" with "lost due".**

**Introduction**

The new van de Velde paper is a better reference for reduced buffering capacity with the effects on alkalinity. See how the results fit with that paper too (rephrase)

**Reply: Thank you for this suggestion. However, it would appear that the paper of van de Velde et al. (2025) is referring to a reduction in the production of ocean alkalinity by re-exposing anoxic sediments to oxygenated water, leading to pyrite oxidation and a removal of alkalinity. This is not the same as changes to the buffering capacity via the remineralisation of organic carbon to dissolved inorganic carbon species. We would, therefore, rather not change the reference. However, we could add a sentence referring to ocean alkalinity destruction (van de Velde et al., 2025) as an additional effect of mobile bottom fishing.**

**Action: Upon reflection, we think that this is an interesting emerging topic but not directly linked to our study. We have therefore decided to not refer to alkalinity destruction.**

45 please specify carbon stocks as this term is easily confused by stock assessments for fish

**Reply: Did you mean line 41? We believe that the term organic carbon stock is known to most researchers working on the same or a similar topic. Besides, we are now giving definitions of relevant parameters in a new chapter (see reply below).**

45 also talk opposing effects of bioturbators on carbon burial

**Reply: We think we already captured that in the following sentence (lines 47 – 49).**

**Action: None**

75 Very good studies but keep in mind that they have the largest amount of variation due to the benthic community response to trawling which is difficult to predict. This isn't a real of Zhang et al., 2024 because it is very difficult to predict benthic community responses but just to note that their results rely on certain assumptions about the benthic community responses which by nature will be quite variable.

**Reply: We feel that discussing caveats of the mentioned papers in the introduction might not be beneficial.**

**Action: None**

70 One of the large uncertainties with Porz (and I think Zhang) et al., 2024 is their assumptions on redistributed fishing effort which is assumed to be adjacent to closed areas in their scenarios. While a reasonable assumption, this is often not the case as fisher behavior is difficult to predict and they often will decide to fish in areas much further away once an area is closed. Honestly, nothing wrong with the text but I do want the authors to keep these caveats in mind when interpreting those studies.

**Reply: No action required.**

85 I know it's not the point of the paper but is there more reasons to believe that trawling will potentially occur in previously unfished grounds? For example, is it already known that there are potentially valuable fish stocks in these areas that fishers may be anticipating? This information isn't essential for the paper but could strengthen arguments about the potential threat of fishing in these grounds.

**Reply: We cite Fauchald et al. (2021), which have demonstrated that bottom fishing has already moved into previously unfished areas in the Barents Sea and that this trend might continue into the future.**

**Action: None.**

**Methods**

125 Most readers won't look back at the more detailed descriptions of

**Reply: This comment seems incomplete.**

**Action: None**

205 Maybe also acknowledge that since the vulnerability metric has a lot to do with the reactivity of OC, much if not most of this is expected to be degrade with or without disturbance. This could also be in the discussion.

**Reply: As mentioned previously, our methodology and the definition of vulnerable organic carbon do not rely on OC reactivity. Instead, we are looking at the bulk effects of mobile bottom fishing on organic carbon. Very labile components like chlorophyll-a might indeed degrade quickly without added anthropogenic disturbance. However, such components do also recover within a year (Tiano et al., 2024) and might thus be a transient part of the organic carbon pool.**

**Action: We have added a sentence to the methods section describing the meta-analysis: "Given the available data, we restrict our study to fishing impacts on bulk organic carbon and make no attempt to account for different reactivity fractions."**

**Results**

290 OC content and densities can be confusing for even biogeochemists sometimes (speaking for myself but probably for others as well). Can you either use more descriptive terminology or just a brief explanation for the distinction please?

**Reply: Given the confusion regarding terminology and the fact that this was picked up by both referees we propose to include a new chapter where we define what is meant by the terms we use in the manuscript (content, density, and stock).**

**Action: We have added a new chapter 2.2 Definitions.**

300 "presumably…" perhaps a bit nit picky but wouldn't this sentence be more appropriate in the discussion. If you do that you can also expand a little on this explanation as well if needed (as it is now, I'm just slightly unsatisfied as a reader).

**Reply: You might be right. However, we think this is too small an aspect to be picked up in the discussion. We suggest instead deleting the word "presumably".**

**Action: We have modified the sentence: "In the low impact scenario, there are also sizable amounts of vulnerable organic carbon to be found in the Skagerrak, since changes in sediment thickness have a relatively higher importance in these mud-dominated environments when changes relating to organic carbon density are relatively low."**

**Discussion**

325 Possibly good to define (again as it is stated earlier) exactly what you mean by vulnerable here as it might be confused with other definitions used in the literature. Readers will need a reminder, I think.

**Reply: Agreed.**

**Action: We have repeated our definition of vulnerable organic carbon in brackets.**

325 Difficult to know exactly if the reduction of OC found in the meta-analysis corresponds to actual carbon lost from the system since much of it is probably just studies that find resuspension and re-allocation of OC. Good to acknowledge this and/or potentially rephrase so readers don't get the wrong idea (if I'm right but perhaps I'm misunderstanding your analysis).

**Reply: The first paragraph of the discussion is simply repeating the main findings to give some context for the discussion. We discuss limitations later on (chapter 4.3), including the issue raised here.**

**Action: None**

345 That's a good point that will be important for other scientists to account for in the future (OC content vs density thing).

**Reply: We agree.**

420 I know that at least in the North Sea, trawling effort has declined in recent decades compared to years past. I'm just curious if there is a similar decline (or not) here and if that is something that should be discussed.

**Reply: We haven't looked into this as part of the study as it was not part of the scope. Additionally, the time series we have spans only 12 years.**

**Action: None**

---

## Author Response (AR2)

Referee #1

I appreciate that the authors have made an effort to clarify some of my concerns. I believe the results are worthy of publication, and I don't need to see the manuscript again. I have only two remaining minor comments that I would recommend the authors to consider.

1. I still have some doubts whether treating bed level change as a "scaling factor" according to Eq. (8) is justifiable, since delta rho_OC and delta d are mechanistically linked. Varying the value of P_crd from 0 to 1 do help to gauge the uncertainty arising from this approach. The new results (Table S2) are much more sensitive to the erosion effect, which the authors acknowledge and have partially adapted the manuscript to reflect. I was therefore puzzled by the newly added lines 372-377 stating that thickness changes play a minor role compared to density changes. Comparing the first and last rows in Table S2 (P_crd=0 vs P_crd=1) indicates to me that the impact of bed changes accounts for roughly half of the effect even in mean and high impact scenarios (as also indicated in l.312-314). I recommend the authors clarify, rephrase or delete this part to avoid confusion.

**Reply:** *We agree that this paragraph does not fully reflect the results. We have therefore reworded it. It now states that thickness changes make a minor contribution to reductions in stocks only when assuming that 87% of the resuspended sediment resettles (Sala et a., 2021). We also point out that this value is associated with considerable uncertainty.*

**Action:** *The paragraph reads now: "In addition, changes to the sediment thickness due to erosion of the top layer of seabed sediment by mobile bottom fishing need to be considered when investigating fishing impacts on organic carbon stocks (eq. 8). Assuming $P_{crd}$ = 0.87 (Sala et al., 2021). Our results indicate that, with the notable exception of the low impact scenario, the estimated reductions in organic carbon stocks are largely due to reductions in organic carbon density, while thickness changes only play a minor role. However, changes to the sediment thickness are sensitive to the choice of $P_{crd}$, which is poorly constrained."*

2. My other comment refers to the newly added subsection "2.2 Definitions". In general, I found it very useful - having definitions to refer to will be useful to the community and may promote more consistency in terminology. However, I found the definition of organic carbon stock as "the mass of organic carbon within a known volume of sediment" unclear, since this definition (a mass within a sediment volume) seems more applicable to the "organic carbon reservoir", reported as mass in kg, and not in kg/m^2 as stock is defined here. I would therefore suggest rephrasing to something like "Organic carbon stock is the mass of organic carbon per area in a defined depth interval

within the sediment." It may also be useful to define "organic carbon reservoir" hereafter, since the terms "stock" and "reservoir" are also often used interchangeably.

**Reply:** *We agree with these comments.*

**Action:** *We have adopted the definition of stocks given by the referee and added a definition for reservoirs:*

*"Organic carbon stock is the mass of organic carbon per area in a defined depth interval within the sediment."*

*"Organic carbon reservoir is the mass of organic carbon in an area and a defined depth interval. Typically, reservoirs are calculated for larger areas such as a sea basin and expressed as Tg C or similar."*

Referee #2

This is an interesting and valuable study. The author evaluates the impacts of demersal fishing on organic carbon stocks and maps the OCS in unfished areas. This work provides important insights for assessing marine carbon storage and serves as a good example for broader application at the global scale. Although I was not one of the first-round reviewers, I have carefully read the revised manuscript, compared it with the previous reviewers' comments and the authors' responses, and I believe that the authors have satisfactorily addressed most of the earlier concerns. Therefore, I recommend that the manuscript be accepted for publication.

**Action:** *None*